



# Radiative Closure Assessment of Retrieved Cloud and Aerosol Properties for the EarthCARE Mission: The ACMB-DF Product

Howard W. Barker[1], Jason N. S. Cole[2], Najda Villefranque[3], Zhipeng Qu[2], Almudena Velázquez Blázquez[4], Carlos Domenech[5], Shannon L. Mason[6], Robin J. Hogan[6]

[1] Environment and Climate Change Canada, Victoria, BC, Canada
[2] Environment and Climate Change Canada, Toronto, ON, Canada
[3] CNRM, Université de Toulouse, Meteo-France, CNRS, Toulouse, France
[4] Royal Meteorological Institute of Belgium, Brussels, Belgium
[5] GMV, Madrid, Spain
[6] European Centre for Medium Range Weather Forecasts, Reading, United Kingdom

*Correspondence to:* Howard W. Barker (howard.barker@canada.ca)

**Abstract**. Measurements made by three instruments aboard the EarthCARE satellite, plus data from auxiliary sources, will be used to synergistically retrieve estimates of cloud and aerosol properties. The ACMB-DF processor consists of a continuous radiative closure assessment of these retrievals and is both described and demonstrated in this study. The closure procedure begins with 3D radiative transfer models (RTMs) acting on retrieved and auxiliary data. These models yield upwelling shortwave and longwave broadband radiances commensurate with measurements made by EarthCARE's multi-angle broadband radiometer (BBR). Measured and modelled radiances are averaged up to "assessment domains", that measure ~21 km along-track by no more than 5 km across-track, centred on the retrieved cross-section of ~1 km profiles, and are then combined, by angular distributions models (ADMs), to produce "effective" upwelling fluxes at the top-of-atmosphere, denoted as $F_{BBR}$ and $F_{RTM}$, respectively. Last, the probability $p_{\Delta\hat{F}}$ of $\left| F_{RTM} - F_{BBR} \right|$ being less than $\Delta\hat{F}$ W m$^{-2}$ is estimated recognizing as many sources of, assumed normally distributed, uncertainties as possible. For historical/programmatic reasons, $\Delta\hat{F}$ is set to 10 W m$^{-2}$, but that might change during EarthCARE's commissioning phase and with Sun angle. The closure process is demonstrated up to calculation of $p_{\Delta\hat{F}}$ using four 400 km-long portions of



one of EarthCARE's test frames for which simulated passive measurements were computed by
3D RTMs. Note that this study, like the ACMB-DF process with real EarthCARE observations,
does not comment explicitly on performance of retrieval algorithms.

# 1. Introduction

The EarthCARE research satellite mission is a collaborative undertaking between the European
Space Agency (ESA) and the Japan Aerospace Exploration Agency (JAXA). Launched on 27-
May-2024 with a payload of cloud-profiling radar (CPR), backscattering lidar (ATLID), passive
multi-spectral imager (MSI), and broadband radiometer (BBR) (see Wehr et al. 2023 for an over-
view). EarthCARE's overarching science goal is to estimate profiles of cloud and aerosol proper-
ties, using CPR, ATLID, and MSI measurements, sufficient well that when operated on by
broadband (BB) radiative transfer (RT) models (RTMs), simulated top-of-atmosphere (TOA) BB
fluxes, for ~100 km$^2$ domains, are accurate to within ±10 W m$^{-2}$ (ESA 2001; Wehr et al. 2023).
*Verifying* this goal, and thus *validating* the scientific and technical choices that led to Earth-
CARE, requires well-defined *closure experiments*. From EarthCARE's outset, the plan has been
to perform a *continuous radiative closure assessment* of its retrieved cloud and aerosol properties
(ESA 2001). Description and demonstration of this procedure is the subject of this paper.

Research satellite missions that retrieve geophysical variables usually involve verification ex-
periments. Ideally, these experiments utilize measurements that contain information not present in
measurements used to make retrievals. Often, they are made from a separate platform, such as
when comparing cloud particle attributes inferred from satellite data to *in situ* samples from
aircraft-mounted sensors that fly within the satellite's field-of-view (e.g., Barker et al. 2008;
Deng et al. 2013; Qu et al. 2018). While *in situ* closure experiments provide invaluable infor-



mation, they are characterized by: i) logistical and interpretive difficulties (e.g., long-term plans that have to work with, and around, meteorological conditions realized over preset periods); ii) limited spatial and temporal sampling spaces (e.g., small sampling volumes covered on short localized flights); and iii) high operating costs that limit spatial, temporal, and sizes of samples.

Alternatives to *in situ* assessments use *ex situ*, or off-site, observations. These include (near-)simultaneous observations of atmospheric volumes made by other remote sensors located on the surface, aircraft, or satellites. In the case of satellites, sensors used for assessment can be on either a satellite of opportunity (e.g., geostationary satellite observations that coincide with those of the research satellite), or the research satellite itself. In the latter case, which is EarthCARE's, geo-

physical quantities retrieved by algorithms that use observations from a subset of the satellite's sensors initialize atmospheric RTMs that predict observations from an exclusive subset of sensors whose observations were *not* used by retrieval algorithms (e.g., Henderson et al. 2013; Ham et al. 2022).

    *Ex situ* closure experiments have advantages and disadvantages relative to their *in situ* coun-

terparts. The greatest advantage is the potential to continuously sample all meteorological conditions encountered throughout a mission. Moreover, while sensors that gather data for assessments incur upfront, and ongoing, data processing costs, they likely serve other purposes, too. On the other hand, the obvious disadvantage is lack of *ground-truth* sampling and the *many-to-one* problem in which key variables (e.g., ice crystal habits and sizes that could be sampled *in situ*) are free

to range over values that lead to indistinguishable responses and, ultimately, weaker assessments. Also, it might be that measurements used to infer geophysical quantities are correlated, to some extent, with measurements used for their assessment, and this weakens assessments, too. Ulti-



mately, the most comprehensive closure assessments of satellite retrievals involve coordinated *ex situ* and *in situ* measurements (e.g., Qu et al. 2018).

In advance of launch, ESA orchestrated a programme to numerically simulate the entire EarthCARE measurement-retrieval-assessment chain of procedures. At the front of this *end-to-end* simulation was production, by a high-resolution numerical weather prediction (NWP) model, of surface-atmosphere conditions for domains that encompass three EarthCARE *frames*, which measure 200 km across-track (i.e., the MSI's swath) by ~6,200 km along-track (Qu et al. 2023b).

These data were then used to approximate synthetic measurements for all four of EarthCARE's sensors (Donovan et al. 2023). These "measurements" were operated on by retrieval algorithms, as summarized in several papers in this special issue, that produce EarthCARE's "best estimate" of cloud and aerosol properties. Retrieved cloud and aerosol properties are then passed to BB RTMs (Cole et al. 2023) that produce, among other quantities, BB TOA radiances that when

compared to their BBR counterparts define the closure assessment and end of the initial versions of EarthCARE's virtual and real processing streams (Eisinger et al. 2023).

     When dealing with synthetic ATLID, CPR, and MSI observations, the most obvious assessment of inferred geophysical variables is to compare them directly to their corresponding NWP model values (see Mason et al. 2024). Clearly, this is not possible for the actual mission whose

purpose is to help improve the NWP model, and others like it, responsible for generating test data in the first place. The present report is consistent with the actual mission in that it stops at description and demonstration of the *ex situ* closure assessment as described above.





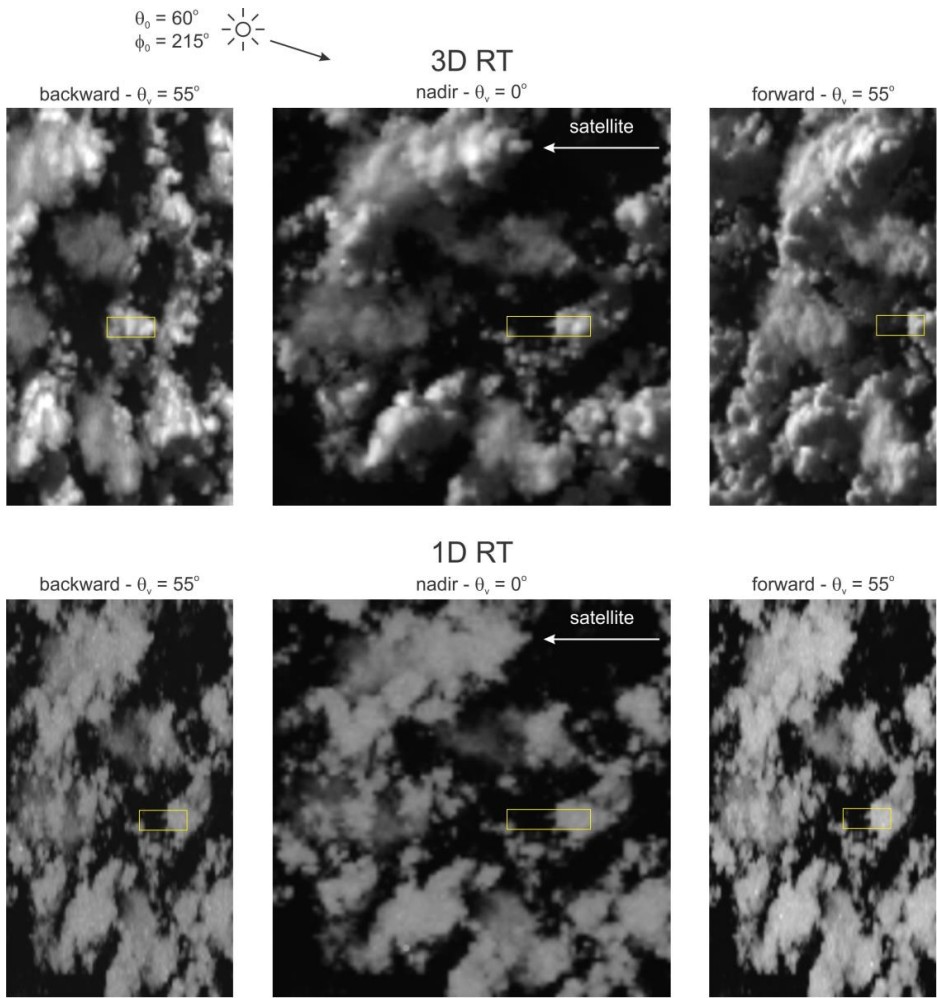

**Figure 1:** Upper panels show BB SW radiances, simulated by a 3D RTM, as observed by the BBR's backward, nadir, and forward pointing telescopes. Viewing zenith angle $\theta_v$ for off-nadir views is $55°$. The entire image is $100 \times 100$ km. Yellow rectangles indicate the size of $5 \times 21$ km assessment domains. Lower panels are the same except radiances were simulated by a 1D RTM. For both simulations solar zenith and azimuth angles were $\theta_0 = 60°$ and $\varphi_0 = 215°$, respectively; $\varphi_0$ is clockwise from north with the satellite tracking due south (see Qu et al. 2023b).

One of EarthCARE's many novelties is use of 3D RTMs, in addition to the usual 1D approximations (Cole et al. 2023). Figure 1 shows shortwave (SW) radiances, computed by 1D and 3D



RTMs, that correspond to the BBR's configuration. In this case, but not all cases, 1D RTM imagery is "flat" (see Barker et al. 2017). On the energetic side, differences between 1D and 3D

RTM heating rates (not shown) can be striking, and so for EarthCARE, BB SW flux profiles will be calculated by 3D RTMs (Cole et al. 2023).

For the current study, 3D RTMs (Villefranque et al. 2019; 2022) were used to simulate MSI and BBR measurements for use in the virtual system, which until now had relied on synthetic radiance observations produced by 1D RTMs (see Donovan et al. 2023; Mason et al. 2023). This

was to demonstrate the need for realistic radiances in both *end-to-end* simulations and the mission proper. Due to computational limitations, 1D and 3D RT radiances were produced for just four domains measuring 400 km along-track by 30 km across-track, at 250 m horizontal resolution.

The following section describes EarthCARE's radiative closure assessment procedure, which defines the so-called ACMB-DF processor. The third section discusses use of synthetic passive

measurements created by 3D, rather than the usual 1D, RTMs. This is followed by application of the closure process to synthetic measurements. A summary and conclusions are presented in the final section.

## 2. EarthCARE's continuous radiative closure experiment

### 2.1. Overview

Geophysical variables retrieved from observations made by EarthCARE's ATLID, CPR, or MSI sensors are referred to as L2 products (see Wehr et al. 2023 and Eisinger et al. 2024 for overview summaries). Products arising from a single sensor's data are designated as L2a, while those from multiple sensors are L2b. L2 products are reported on all or part of the Joint Standard Grid (JSG),





which has horizontal resolution of ~1 km and, looking forward along the satellite's motion vec-

tor, extends across-track 35 km to the right and 115 km to the left; the asymmetry helps reduce

complications that arise from sunglint. Vertically-resolved L2 variables are on 0.1 km-thick

layers, extend from surface to 20 km, and form the L2-plane. The focus of radiative closure as-

sessments is on L2b profiles of cloud and aerosol properties.

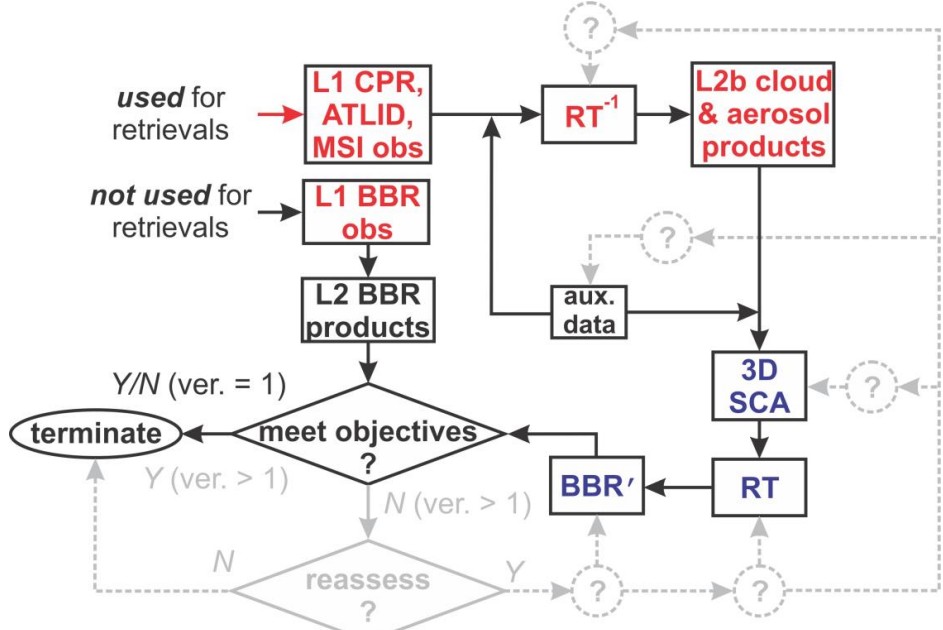

**Figure 2:** Flowchart showing EarthCARE's radiative closure assessment programme. Version 1
(ver. = 1) represents EarthCARE's initial processing plan. It terminates unconditionally after
comparing modelled to measured BBR quantities whilst reporting the likelihood of their differ-
ence being within $\pm 10$ W m$^{-2}$. For subsequent processings (ver. > 1), it is expected that if mod-
elled and measured BBR quantities compare unsatisfactorily, potentially all steps in the pro-
cessing chain will be interrogated and adjusted until some level of agreement is reached.

Figure 2 summarizes the flow of products leading to, and including, EarthCARE's *ex situ* radi-

ative closure experiment. It begins with L2b variables and auxiliary information, from model

analysis (see Eisinger et al. 2023) and climatological statistics (Qu et al. 2023b), being used by



the 3D Scene Construction Algorithm (SCA) (Barker et al. 2011; Qu et al. 2023a). Using MSI

radiances, the SCA associates an off-nadir JSG pixel with its closest matching nadir pixel. L2b

profiles, and surface properties, associated with the donor nadir column get replicated at the off-

nadir recipient to form a 3D surface-atmosphere system around, and consisting entirely of data in,

the L2-plane.

Information from the SCA gets ingested into various forward radiative transfer models (Cole

et al. 2023) that predict profiles of BB radiative fluxes as well as upwelling BB radiances at TOA

that are commensurate with BBR observations. The essence of the closure assessment, which

marks the end of version 1 of EarthCARE's production chain, is comparison of TOA *effective

fluxes* that derive from modelled and measured radiances averaged over assessment domains

(AD). Following Qu et al.'s (2023a) notation, assessment domains consist of $n_{assess}$ JSG pixels

along-track with across-track half-widths of $m_{assess}$ JSG pixels, for a total of $(2m_{assess}+1)n_{assess}$

JSG pixels. The current plan (Qu et al. 2023a) is $n_{assess} = 21$ and $m_{assess} = 2$, so that assessment

domains will measure ⧄ $5 \times 21$ km.

## 2.2. Closure assessment variable

The most direct closure assessments use the BBR's three directional radiances. Nadir BBR radi-

ances, by themselves, provide weak closure tests, for as shown elsewhere (e.g., Barker et al.

2014), both SW and LW BB nadir radiances can be correlated well with MSI radiances that are

used by some L2 retrieval algorithms (e.g., Mason et al. 2023). Off-nadir BBR radiances have

viewing geometries that differ markedly from all other EarthCARE sensors, are usually much less

correlated with MSI nadir radiances than are BB nadir radiances, and so have the potential to



provide stringent radiative closure assessments. There is always, however, the possibility that substantial fractions of photons that constitute off-nadir BBR radiances have trajectories that depend much on atmospheric attenuators and surfaces outside the AD. This happens when cloud and aerosol occur between the BBR and AD, and when bright clouds or surfaces backlight an

AD. In extreme cases, off-nadir radiances might say very little about the quality of retrievals within the AD (Barker et al. 2015; Tornow et al. 2015).

Another issue with direct use of radiances is that it breaks with EarthCARE's long-held science goal that states explicitly that retrieval quality be gauged in terms of W m$^{-2}$ (ESA 2001; Illingworth et al. 2015; Wehr et al. 2023). To abide by this, the obvious approach is to compare

TOA fluxes predicted by ACM-RT's RTMs to corresponding values obtained by EarthCARE's Angular Distribution Models (ADMs) (Velázquez Blázquez et al. 2024a), which for the SW are based on CERES ADMs (Loeb et al. 2005; Domenech and Wehr 2011), and for the LW on the operational GERB LW flux estimation (Clerbaux et al. 2003a,b).  Once outside the idealized world of 1D RT, however, defining TOA fluxes for $5 \times 21$ km, or smaller, atmospheric columns

is fraught with ambiguity and potentially large, and difficult to quantify, uncertainties (cf. Kato and Loeb 2005).

For these reasons, it was decided that the most well-defined, reliable, and programmatically satisfying way to perform radiative closure assessments is to transform "both" BBR measured and ACM-RT simulated TOA BB radiances into "effective fluxes" via EarthCARE's ADMs. The

attraction of using $F_{RTM} - F_{BBR}$, where $F_{RTM}$ and $F_{BBR}$ are effective fluxes derived from either an RTM's or the BBR's radiances, for the closure assessment variable is that it largely sidesteps uncertainties associated with instantaneous application of ADMs and complications around exact definition of TOA fluxes. It does mean, however, that true "fluxes" never enter EarthCARE's





closure assessments and that potential issues associated with the use of off-nadir radiances, as
mentioned above, go unaddressed (at least for EarthCARE's initial processing).

Following Velázquez Blázquez et al. (2024a), longwave effective fluxes are defined as

$$\mathsf{F}_{\mathrm{BBR}} = \frac{1-\alpha}{2}\Big[F_{\mathrm{BBR}}(1) + F_{\mathrm{BBR}}(3)\Big] + \alpha F_{\mathrm{BBR}}(2), \tag{1}$$

where flux estimates from each telescope are

$$F_{\mathrm{BBR}}(i) = \frac{\pi L_{\mathrm{BBR}}(i)}{R(i)}, \tag{2}$$

$\alpha = 1/3$, $L_{\mathrm{BBR}}(i)$ are unfiltered BBR radiances (W m$^{-2}$ sr$^{-1}$), in which $I = 1, 2$, and 3 correspond
to forward, nadir, and backward views, respectively, and $R(i)$ are parametrized anisotropic
factors that depend on MSI brightness temperatures. Model-generated counterparts of (1), desig-
nated as $\mathsf{F}_{\mathrm{RTM}}$, are computed the same way except that Monte Carlo estimated radiances $L_{\mathrm{RTM}}$
replace $L_{\mathrm{BBR}}$ in (2) (see Cole et al. 2023).

Shortwave effective fluxes are more difficult to define than $\mathsf{F}_{\mathrm{BBR}}$ because of pronounced ani-
sotropy. Following Velázquez Blázquez et al. (2024a), ADM-based fluxes derived from the nadir,
aft, and fore views are combined as

$$\mathsf{F}_{\mathrm{BBR}} = \Bigg[\sum_{i=1}^{3} \frac{\delta(i)}{\varepsilon_{F_{\mathrm{BBR}}(i)}\pi\varepsilon_{R(i)}}\Bigg]\Bigg[\sum_{i=1}^{3} \frac{\delta(i)F_{\mathrm{BBR}}(i)}{\varepsilon_{F_{\mathrm{BBR}}(i)}\pi\varepsilon_{R(i)}}\Bigg], \tag{3}$$

where $F_{\mathrm{BBR}}(i)$ are as in (2) but the anisotropic factors for each view are obtained from an artifi-
cial neural network trained with surface and atmospheric analysis data, and $\varepsilon_{F_{\mathrm{BBR}}(i)}$ and $\pi\varepsilon_{R(i)}$ are



flux uncertainties arising from the ADMs and the BBR unfiltered radiance estimation (Velázquez Blázquez et al. (2024b), respectively. When all $F_{\text{BBR}}(i)$ agree to within $\pm 10\%$, $\delta(i)=1$ for all $i$. When two $F_{\text{BBR}}(i)$ agree to within $\pm 10\%$, each uses $\delta = 1$ with the outlier getting $\delta = 0$. If all $F_{\text{ADM}}(i)$ differ by more than 10%, only the smallest $\varepsilon_{F_{\text{BBR}}(i)}\pi\varepsilon_{R(i)}$ uses $\delta = 1$. When computing

$F_{\text{RTM}}$ with (3), 3D Monte Carlo RTM radiances $L_{\text{RTM}}$ (Cole et al. 2023) replace $L_{\text{BBR}}$, and Monte Carlo radiances uncertainties $\varepsilon_{R(i)}$ replace BBR measurement uncertainties.

An optional approach is to eliminate radiance uncertainty from (3) by stochastically sampling $L_{\text{BBR}}(i)$ and $L_{\text{RTM}}(i)$ and producing distributions of $F_{\text{BBR}}$ and $F_{\text{RTM}}$. This has the potential to sample multiple combinations of $\delta(i)$ and hence substantially broaden distributions of plausible

$F_{\text{BBR}}$ and $F_{\text{RTM}}$. This detail will be explored during EarthCARE's commissioning phase.

When estimating effective flux based on the BBR's three views, an ever-present issue is co-registration of radiances to ensure that they correspond to the AD defined at nadir. In general, this requires dynamic specification of an altitude that corresponds to where the majority of photons received by the telescopes begin their final upward trajectories. For clear-skies, this is (close to)

Earth's surface; especially for SW radiation. For cloudy-skies, however, this could be anywhere from surface to cloudtop, and cloudtop might be outside the AD (see Barker et al. 2014).

## 2.3. Closure assessment metric

We assume that "best estimates" of $F_{\text{BBR}}$ and $F_{\text{RTM}}$, averaged over $D$, are mean values of underlying Gaussian distributions $N\left(F_{\text{BBR}}, \sigma^2_{F_{\text{BBR}}}\right)$ and $N\left(F_{\text{RTM}}, \sigma^2_{F_{\text{RTM}}}\right)$, where $\sigma^2_{F_{\text{BBR}}}$ and $\sigma^2_{F_{\text{RTM}}}$



are respective standard deviations and taken to be "uncertainties". Although some key input variables for ACM-RT's RTMs will have estimated uncertainties, computational limitations and time constraints (see Cole et al. 2023) mean that many contributions to $\sigma^2_{F_{RTM}}$ will be neglected.

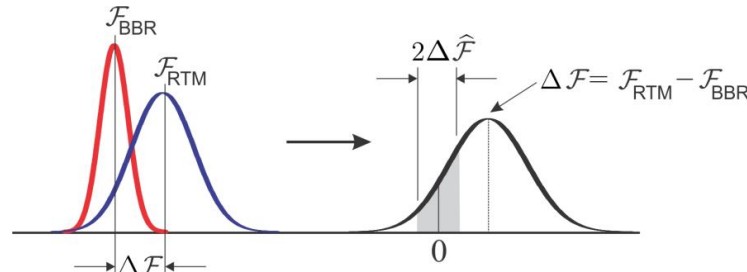

**Figure 3:** Schematic illustrating (left) assumed normalized Gaussian distributions of measured

$N\left(F_{BBR}, \sigma^2_{F_{BBR}}\right)$ (red) and modelled $N\left(F_{RTM}, \sigma^2_{F_{RTM}}\right)$ (blue) fluxes and the resulting (right) Gauss-ian distribution of their difference $N\left(\Delta F, \sigma^2_p\right)$. Area of the shaded region is the probability that $F_{BBR}$ and $F_{RTM}$ differ by less than $\pm\Delta\hat{F}$.

This also pertains to auxiliary variables, not inferred from EarthCARE retrievals, such as surface

optical properties, and temperature and moisture profiles. Nevertheless, define

$$\Delta F = F_{RTM} - F_{BBR}, \tag{4}$$

and assume that pooled uncertainty can be approximated simply as

$$\sigma^2_p \approx \sigma^2_{F_{BBR}} + \sigma^2_{F_{RTM}}. \tag{5}$$

Therefore, estimated probability of $\left|\Delta F\right| \le \Delta\hat{F}$ is



$$p_{\Delta\hat{F}} = \frac{1}{\sigma_p \sqrt{2\pi}} \int_{-\Delta F}^{\Delta\hat{F}} \exp\left[-\frac{(x - \Delta F)^2}{2\sigma_p^2}\right] dx$$

$$= \frac{1}{2}\left[erf\left(\frac{\Delta\hat{F} - \Delta F}{\sqrt{2}\sigma_p}\right) - erf\left(\frac{-\Delta\hat{F} - \Delta F}{\sqrt{2}\sigma_p}\right)\right], \tag{6}$$


where $erf(\cdots)$ is the error function. The quantity $p_{\Delta\hat{F}}$ provides a succinct indication of the likelihood that L2 products, and to a lesser extents auxiliary data and SCA performance, have been retrieved well enough to be designated as having satisfied the mission's goal of $\pm\Delta\hat{F}$, at the scale of the AD. Figure 3 illustrates this schematically.

The tacit assumption, thus far, has been that use of $\Delta\hat{F} = 10$ W m$^{-2}$, EarthCARE's goal, applies everywhere, all the time; it has never been specified if it applies to SW and LW radiation separately, or to their sum. While it is reasonable be say "everywhere, all the time" for LW radiation, it is not for SW fluxes, where aiming for $|\Delta F| \leq 10$ W m$^{-2}$ at small $\theta_0$ is much more demanding than at large $\theta_0$. What has been settled on for SW radiation is to replace $\Delta\hat{F}$ in the

above equations with $\Delta\hat{F}\mu_0 / \langle\mu_0\rangle$, where $\mu_0$ is local value of $\cos\theta_0$ and $\langle\mu_0\rangle$ is arithmetic mean of $\mu_0$ for the portion of EarthCARE's orbit with $\mu_0 > 0$. For simplicity, $\Delta\hat{F}$ for both SW and LW do not depend on surface or atmospheric conditions.

## 3. On the use of 3D RTMs to simulate observed radiances

Simulated radiometric observations produced by 1D RTMs are often used for development and

testing of cloud and aerosol retrieval algorithms (see Donovan et al. 2023; and many other papers in this special issue). For EarthCARE, 1D RTMs were needed, because of computational burden,





to simulate observations for three large test frames (Qu et al. 2023b) at spectral and spatial resolutions high enough to capture radiometer filter functions and spectral unfiltering (Velázquez Blázquez et al. 2024b). A better approximation of real conditions is achieved, however, when 3D

RTMs are used to simulate radiances. To demonstrate the closure assessment process, all passive radiances were computed by 3D RTMs (Villefranque et al. 2019) at horizontal grid-spacing of $\Delta x = 0.25$ km, which is the resolution of test frame data, for four select $\sim 400 \times 30$ km portions of the *Hawaii* frame; setting $\Delta x \to \infty$ affects 1D RT conditions commensurate with all other tests reported in this special issue.

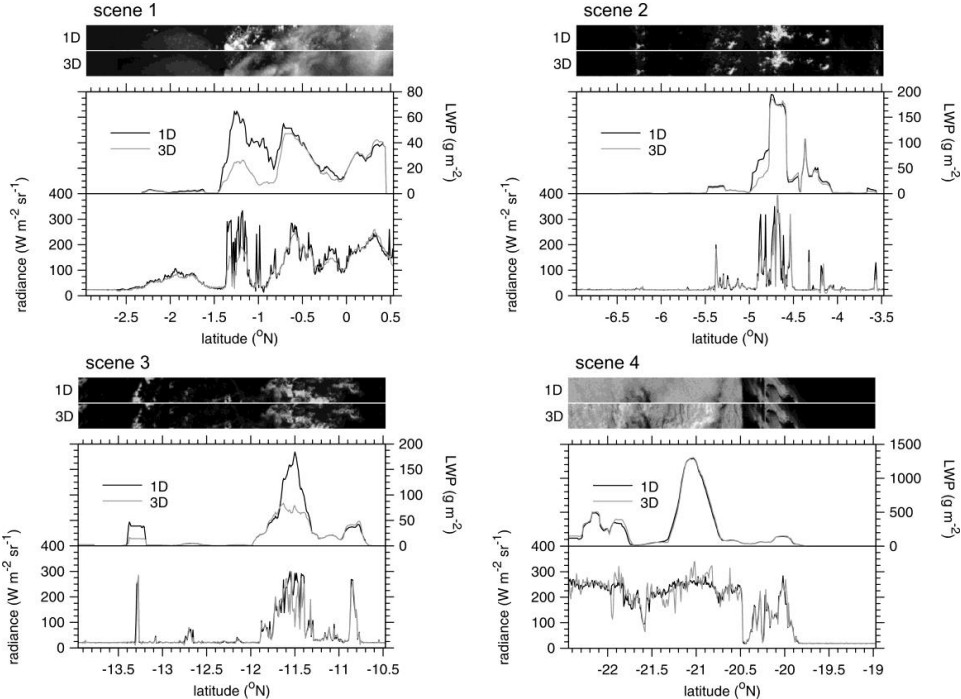


**Figure 4:** For each ⬚ $400 \times 30$ km scene, sampled from the Hawaii frame, top images are MSI 0.67 µm nadir radiances computed using 1D and 3D RTMs. Line plots show 1D and 3D radiances along the centres of the images and corresponding cloud LWP inferred by the CAPTIVATE (ACM-CAP) retrieval algorithm (Mason et al. 2023) when constrained by 1D or 3D MSI radiances. The mean solar zenith angles $\theta_0$ for scenes 1, 2, 3, and 4, are 37º, 40º, 45º, and 51º, respectively.


tively.





As use of 3D RTMs to simulate observed and modelled radiances represents a marked departure from all other reports in this issue, impacts due to this change are presented briefly here. It should be noted, however, that the point of this section, and indeed the entire paper, is not to explain, or examine in detail, retrieval algorithm performances, but rather focus on the closure methodology.

Figure 4 shows the impact of constraining CAPTIVATE's synergistic retrieval algorithm (ACM-CAP; Mason et al. 2023) with MSI radiances simulated by either a 1D or 3D RTM. Scene 1 is covered by ice cloud with ice water path (IWP) generally larger than 20 g m$^{-2}$, save for 1°S to 1.5°S where the nadir cross-section is almost ice-free. Nearby ice clouds, however, cast shadows onto low-level liquid clouds for 3D RT but not for 1D RT. Therefore, when radiances are based on 3D RT, liquid clouds appear to CAPTIVATE, which assumes 1D RT, to be too thin. The other scene with upper-level ice cloud is 4, which has widespread IWP of ~400 g m$^{-2}$ near 21°S and 65 g m$^{-2}$ near 22°S. In this case, irradiance onto low liquid clouds depends little on the type of RTM, and so LWP values are very similar.

In contrast, scenes 2 and 3 have almost no ice cloud so differences in retrieved LWPs stem from either side illumination or shadowing. Generally, 3D RT values are very close to, or less than, their 1D counterparts implying shadowing and entrapment of photons (cf. Hogan et al. 2019) are of some importance. These results illustrate the need to assess retrieval algorithms with MSI radiances simulated by 3D RTMs, for they provide better indications of what to expect once operating with real data.

For demonstration of the radiative closure assessment in the following section, BBR radiances simulated by 3D RTMs, at $\Delta x = 0.25$ km, were averaged up to assessment domains that measure





either 5 km across-track by 21 km along-track, denoted as $AD_{5x21}$, or 1×21 km, denoted as $AD_{1x21}$. The former represent EarthCARE's default domains that are centred on cross-sections of retrieved geophysical variables and include small areas on both sides that are filled by the SCA (Barker et al. 2011; Qu et al. 2023a). This eases the burden of alignment of measurements, but also factors into assessments of retrievals results from the SCA. While use of $AD_{1x21}$ restricts

295  closure assessments to retrieved cross-sections, which limits the SCA's role to facilitation of, in 3D RTMs, across-track horizontal transport of photons in and out of $AD_{1x21}$, assessment credibility might be compromised by requiring BBR measurements to perform outside of its design specifications? Use of $AD_{5x21}$ and $AD_{1x21}$ this will be explored during EarthCARE's commissioning phase. Note that while maximum across-track size of an assessment domain is 17 km (for

300  details, see Velázquez Blázquez et al. 2024), that would put far too much emphasis on performance of the SCA.

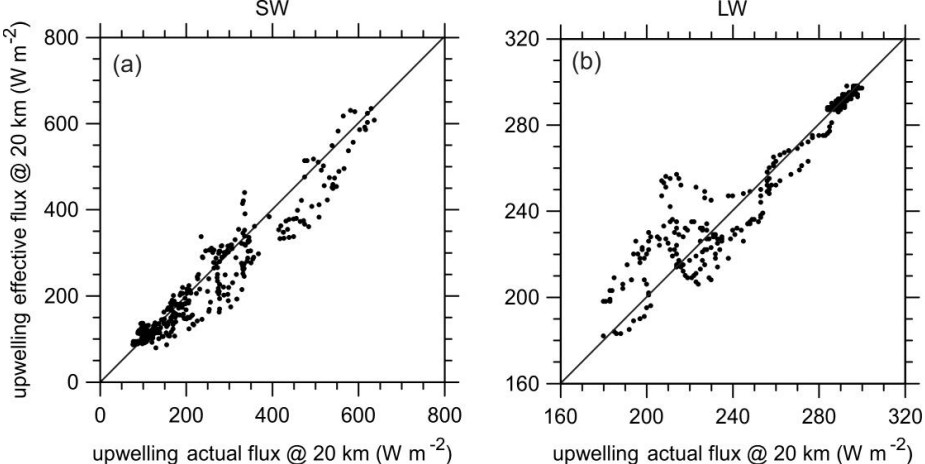

**Figure 5:** (a) Upwelling effective SW flux at 20 km altitude predicted by (3) using radiances at three BBR angles against their actual (i.e., hemispheric integrated) counterparts for all $AD_{5x21}$ in

305  the four scenes. (b) As in (a) except these are LW quantities.



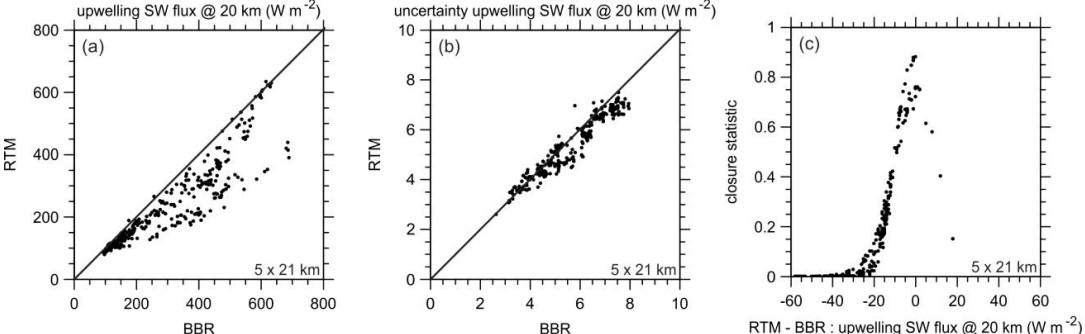

**Figure 6:** (a) Upwelling effective SW flux at 20 km altitude predicted by (3) using radiances at three BBR angles based on cloud properties inferred by ACM-CAP against their counterparts based on input cloud properties produced by GEM for all assessment domains $AD_{5x21}$; the former represent quantities that will come from the ACM-RT process, while the latter represent quantities that will come from the BMA-FLX process using BBR observations. (b) Effective SW flux uncertainties that correspond to values in (a). (c) Closure assessment metric $p_{\Delta\hat{F}}$ using values in (a) and (b) assuming $\Delta\hat{F} = 10\mu_0 / \langle\mu_0\rangle$ W m$^{-2}$.

# 4. Results

It is instructive to first check on relations between *true* TOA broadband fluxes predicted directly by 3D RTMs and corresponding $F_{RTM}$ based on their simultaneously estimated radiances (see (1) and (3)). Figure 5 shows these comparisons for SW and LW fluxes for all $AD_{5x21}$ in the four scenes; results for $AD_{1x21}$ are very similar and not shown. Due to concerns and ambiguities discussed in section 2.2, in addition to $F_{RTM}$ being based on just three radiances, these quantities are not expected to agree perfectly. For cloudless domains with small reflectances and the most reflective overcast domains, values of $F_{RTM}$ agree quite well with their true counterparts. Random deviations are more apparent, often exceeding $\pm100$ W m$^{-2}$, with just a weak tendency for $F_{RTM}$ to underestimate true flux.



As expected, LW values of $F_{RTM}$ agree much better with their true counterparts. This is almost certainly because it is simpler to estimate fluxes based on few radiances for LW radiation than for SW. As alluded to above, the important point here is that because "effective fluxes" for the RTMs are arrived at the same way as they are for BBR measurements, they should provide solid closure assessments that remain as true as possible to ESA's overarching science requirements and objectives.

Figure 6a shows $F_{BBR}$ against $F_{RTM}$ for SW radiation and all $AD_{1x21}$. For mostly cloudless conditions, agreement is very good, but as reflectance, and thus cloudiness, increases, $F_{RTM}$ becomes increasingly less than $F_{BBR}$ and appears to bifurcate for the most reflective domains with one branch being very poor agreement and the other excellent. At this stage, there are no simple and obvious relations between $F_{RTM} - F_{BBR}$ and cloud properties. The objective here, however, was just to demonstrate the methodology and role of the closure process, not to explain retrieval algorithm performance; that is for the commissioning phase. Nevertheless, *Figure 6*b shows approximate uncertainties of RTM and BBR fluxes to be used in (6); the former stem from Monte Carlo noise, while the latter from errors relative to CERES ADM values. The fact that they are of very comparable magnitude is purely coincidental given the 300,000 photons per domain used in the Monte Carlo RTM. What is clear is that uncertainties are relatively small thanks to the use of effective fluxes that sidestep ADM errors, which can be large for individual domains (e.g., Loeb et al. 2007).

Figure 6c shows $p_{\Delta\hat{F}}$, which is the end of the first step of the closure assessment, based on values shown in Figure 6a and Figure 6b. Given the similar flux uncertainties, the assumed Gaussian character of $p_{\Delta\hat{F}}$ is apparent here as a function of $\Delta F$. Moreover, the vast majority of





D$_{5\times21}$ have $p_{\Delta\hat{F}} < 0.2$. Even when using $10\mu_0 / \langle\mu_0\rangle$ W m$^{-2}$, $p_{\Delta\hat{F}}$ only reaches ~0.9 on account of

$\sigma_p^2$ often approaching $10\mu_0 / \langle\mu_0\rangle$. This showing differ from that in Illingworth et al. (2015)

where many domains showed $p_{\Delta\hat{F}} > 0.75$. The likely explanation for this disparity is that the

case in Illingworth et al., 2015 had greater consistency between input and inferred geophysical

properties. Namely, inputs were already constrained by CERES radiances, whereas in the present

case inputs were defined upfront, and retrievals operated freely as they will with real observa-

tions.


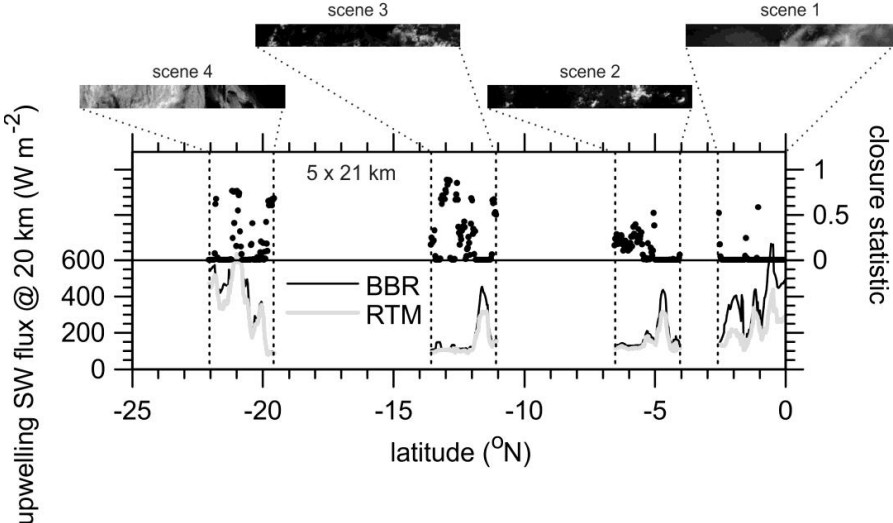

**Figure 7:** Line plot shows values of upwelling effective SW fluxes at 20 km altitude that are
shown in Figure 6a. Upper portion shows $p_{\Delta\hat{F}}$ that are shown in Figure 6c for the four scenes as
functions of latitude. MSI channel 1 images, from the 3D RTM, are shown for reference.





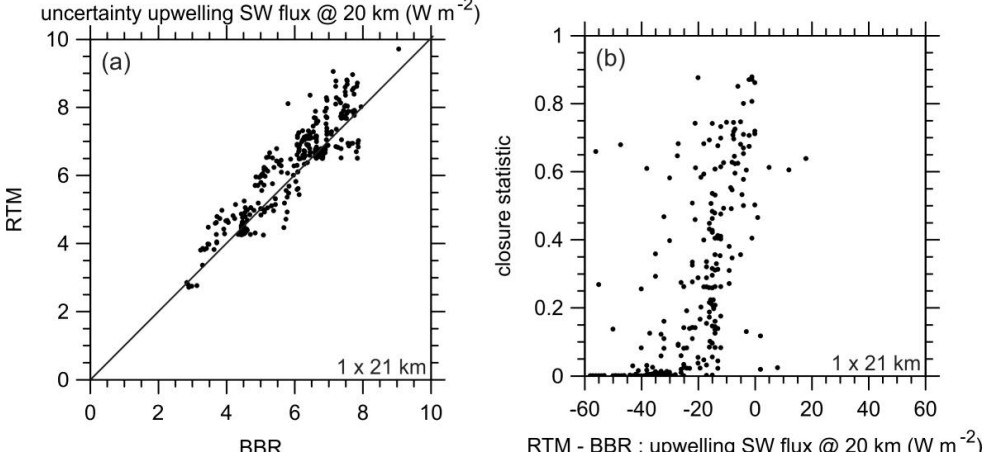

**Figure 8:** (a) and (b) are as in Figure 6b and Figure 6c except these are for domains $AD_{1x21}$ that include just the retrieved cross-section.


Figure 7 shows the values seen in Figure 6a and c as a function of latitude along with MSI channel 1 nadir radiances. Note that ACM-CAP retrievals operated on these radiances, in addition to MSI thermal radiances, that were simulated by a 3D RTM. With this small sample it is difficult to discern trends that are worthy of discussion.

Figure 8 shows effective flux uncertainties for BBR and RTM, $\sigma^2_{F_{BBR}}$ and $\sigma^2_{F_{RTM}}$, and $p_{\Delta F}$ for $AD_{1x21}$. In general, Monte Carlo flux uncertainties are larger than they are for $AD_{5x21}$ because the number of injected photons into $AD_{1x21}$ and their buffer-zones is often significantly less than into $AD_{5x21}$ and their buffer-zones (see Cole et al. 2023). The result is a less stringent closure assessment and larger $p_{\Delta F}$, to the point of $p_{\Delta F} > 0.5$ for some instances of $\left| \Delta F \right| > 50$ W m$^{-2}$. Note, too, that for $AD_{1x21}$, errors in RTM fluxes that arise from the SCA, as small as they usually are,

do not, unlike for for $AD_{5x21}$, enter explicitly into the assessment, as the assessment domain has





collapsed to the retrieved cross-section with the SCA providing boundary conditions only. Nevertheless, it is wise to keep $\sigma^2_{F_{RTM}}$ as small as resources allow.

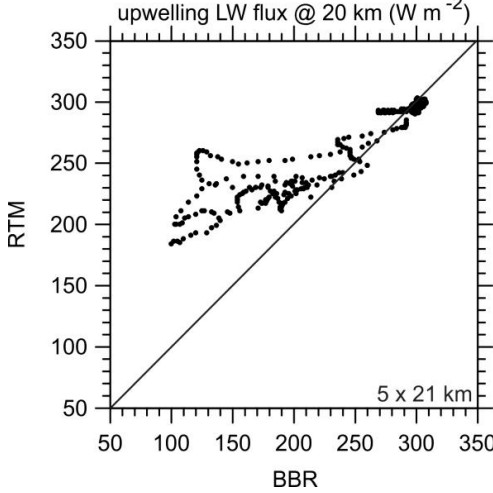

 **Figure 9:** As in Figure 6a but this is for LW effective fluxes.

Figure 9 shows $F_{BBR}$ against $F_{RTM}$ for LW radiation and all $AD_{5x21}$. As in Figure 6a, they agree nicely when fluxes are large, which is for very thin cloud and clear-sky. Surprisingly, however, as clouds become thick or more abundant, and as fluxes decrease, RTM radiances resulting

from retrievals increasingly exceed BBR radiances. This is surprising despite cloudtop altitudes being placed well via active sensor observations. Nevertheless, differences can be traced to underestimation of high ice cloud water contents.

LW flux uncertainties are often $< 0.5$ W m$^{-2}$, which are much less than those for SW fluxes. As Figure 10 shows, the result is that $p_{\Delta \hat{F}}$ tend to bounce between 0 and 1; the former when cold

cloud are missing from retrievals, and the latter when only warm low clouds are present (i.e., for the two centre scenes). Clearly there are issues here that must be resolved. While there is the



potential luxury here to check retrieved cloud properties against input values, the assessment was

cut short to better resemble use of real observations where this luxury does not exist.

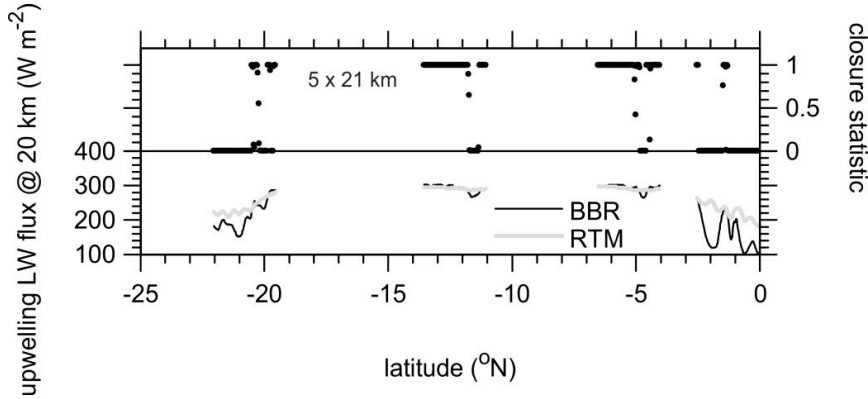

**Figure 10:** As in Figure 7 but this is for LW effective fluxes.

## 5. Summary and discussion

This paper described and demonstrated EarthCARE's planned radiative closure assessment procedure. The assessment's primary objective is to help retrieval algorithm developers diagnose and improve their algorithms during the mission. Second, it is intended to guide users of EarthCARE products who may wish to limit analyses according to performance in the closure assessment. It is important to stress that the intention of this report was *not* to diagnose or assess the quality of retrieval algorithms; that is taking place in other studies, including several in this Special issue, and will unfold in earnest after launch.

From early in EarthCARE's development, a continuous radiative closure assessment was planned (ESA 2001). The procedure is conceptually simple: geophysical properties inferred from EarthCARE observations and auxiliary data sources, get acted on by broadband radiative transfer



models (RTMs), and their results get compared to (near-)simultaneous measurements made by EarthCARE's broadband radiometer (BBR). Crucially, BBR observations are not used by retriev-

al algorithms. The idea is that when modelled and measured quantities appear *highly likely* to differ by less than $\Delta\hat{F}$, as articulated in the mission's goal (ESA 2001; Illingworth et al. 2015; Wehr et al. 2023), the retrievals (plus auxiliary input data) are deemed to be *a success*. When, on the other hand, their difference Is *too likely* to exceed $\Delta\hat{F}$, developers or users may wish to view the retrievals as suspect and in need of addressing, somewhere in the chain (Eisinger et al. 2023),

before they pass muster. In this report, the process of assigning a quantitative measure of retrieval performance was explained and demonstrated.

As with most other reports in this special issue, the closure procedure was demonstrated using synthetic EarthCARE observations made by applying a suite of models to simulated atmosphere-surface conditions (Qu et al. 2023b). An important point of departure from all other reports,

however, was use of MSI radiances, by retrieval algorithms, and BBR radiances, for flux estimation, that were computed by 3D RTMs; all other reports employed 1D RTM results (Donovan et al. 2023). Obviously, 3D RTMs produce synthetic measurements that better represent real measurements. That said, note again that retrieval performance as a function of 1D v. 3D RTM-based synthetic observations were not reported here but rather in forthcoming studies.

The cleanest way to perform a closure assessment is to limit it to just observations. In this case, that means BBR radiances. From the outset, however, EarthCARE's goal has been to make cloud and aerosol retrievals that are accurate enough that when used in RTMs, predicted top-of-atmosphere (TOA) fluxes differ from their "observed" counterparts by less than $\Delta\hat{F} = 10$ W m$^{-2}$. To remain consistent with this publicly stated goal (ESA 2001; Illingworth et al. 2015; Wehr et

al. 2023), it was decided that the most reliable and inclusive closure methodology would be to



transform RTM radiances into "effective fluxes" the same ways that EarthCARE's Angular Distribution Models (ADMs) transform BBR radiances (Velázquez Blázquez et al. 2024a). This approach is attractive in that it sidesteps the potentially overwhelming uncertainties associated with single applications of ADMs to small domains and subsequent comparison to ill-defined TOA fluxes produced by 3D RTMs.

Nevertheless, regardless of the variable(s) used, a closure assessment's strength depends, conditionally, on state variables needed by RTMs (e.g., temperature profiles and surface properties). These variables are likely to be outside the purview of mission retrievals, can be highly uncertain, and thus have the potential to seriously compromise the quality and utility of assessments. As the mission unfolds, much attention will be given to quantifying as many uncertainties as possible. Uncertainties associated with EarthCARE's effective fluxes come from: minor issues associated with BBR radiances (Velázquez Blázquez et al. 2024b); known but approximate errors associated with EarthCARE's ADM (Velázquez Blázquez et al. 2024a), and Monte Carlo noise from EarthCARE's 3D RTMs (Cole et al. 2023). Note that large pooled uncertainties (see (5)) for ADM- and RTM-based values of effective flux can appear to improve an assessment by increasing the probability $p_{\Delta\hat{F}}$ that two fluxes differ by less than $\Delta\hat{F}$. Likewise, if uncertainties are underestimated, or worse neglected, retrievals will appear as failures regardless of how little their effective fluxes differ. In other words, in addition to reported likelihoods of effective fluxes differing by less than $\Delta\hat{F}$, users should pay attention to various uncertainties. Particularly insidious are scenarios in which RTMs operate on erroneous, yet assumed to be perfect, inputs, such as surface temperature, albedo, and BRDF that unwittingly yield contaminated TOA radiances, and ultimately values of $p_{\Delta\hat{F}}$ that could say little about the quality of retrievals.




On a related point, under some conditions cloud evolution and advection can be notable over ~3 minutes, which is the length of time between forward and backward BBR viewings. Given the observations at hand, it is almost impossible to reliably quantify how such conditional changes impact estimates of both BBR and RTM effective fluxes. Again, this has the potential to compromise the integrity of closure assessments. Thus far, all simulations of EarthCARE observations have neglected this detail.

Since at least Tornow et al. (2018), it has been the intention to perform radiative closure assessments on domains that measure 5 km across-track by 21 km along-track. Cloud and aerosol properties are, however, retrieved for nadir columns that are ~1 km wide. Thus, ~80% of each 21 km-long assessment domain relies directly on the performance of the Scene Construction Algorithm (SCA) (Barker et al. 2011; Qu et al. 2023a). This is not ideal and gives rise to minor bias errors (see Barker et al. 2014) that can be estimated from MSI radiances (for the tests reported on here, these errors were very minor and not shown). The benefit of 5 km-wide domains is that BBR radiances are commensurate with design specs (e.g., Velázquez Blázquez and Clerbaux 2010). There is the possibility, as shown here, to limit assessment domains to include just the retrieved cross-section, thereby relegating the SCA to purveyor of boundary conditions that enable handling of across-track photon transport by the 3D RTMs. This will, however, stress the performance of the BBR and instrument co-registration. The final decision on domain size, and myriad other issues, will be made during the commissioning phase with the aid of tentatively planned *in situ* closure experiments.



## Author contributions

HWB drafted the manuscript and developed the methodology presented in this manuscript. JNSC, ZQ, and MK developed several pieces of software that produced data used here. NV developed 3D RT codes and generated all observations based on 3D RTM simulations. AV and CD developed ADM algorithms that are used to produce effective fluxes. SM and RH developed the cloud retrieval algorithm.

## Competing interests

The authors have no competing interests to declare.

## Acknowledgements

We are especially indebted to Dr. Tobias Wehr, who passed away on 1-Feb-2023, for his unwavering support and encouragement over many years of work. We also wish to thank Michael Eisinger and all EarthCARE algorithm development team members for their ongoing support, especially Meriem Kacimi (ECCC) and Edward Baudrez (RMIB) for technical help with this study.

## Financial support

This study is supported by Clouds, Aerosol, Radiation - Development of INtegrated ALgorithms (CARDINAL) for the EarthCARE Mission.



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
