# Peer review of "Radiative Closure Assessment of Retrieved Cloud and Aerosol Properties for the EarthCARE Mission: The ACMB-DF Product"

_EGUsphere, 2024_

## Author Response (AR1)

Responses to Reviewers' comments on: ***Radiative Closure Assessment of Retrieved Cloud and Aerosol Properties for the EarthCARE Mission: The ACMB-DF Product*** by Barker et al. (EGUSPHERE-2024-1651)

**Reviewer 1**

**Reviewer**: On page 6, the authors state that this study is to demonstrate end-to-end simulations with realistic radiances. To me, end-to-end simulations mean that 1) use modeled cloud fields and compute 3D radiative transfer to model BBR and MSI radiances (true radiances). Input these radiances to retrieval algorithms to retrieve cloud and aerosol properties. Use retrieved cloud and aerosol properties combined with a scene construction algorithm to construct 3D cloud fields. Compute TOA radiances using these retrieved cloud fields (retrieved radiances). Apply angular distribution models to both true and retrieved radiances. Assess the difference of true irradiances and retrieved irradiances. I do not think that this is done in this study.

**Reply**: We agree with the Reviewer's definition of "end-to-end" simulation in the context of EarthCARE, as do we agree that, as stated on pg. 6, it was executed, in stages, by many EarthCARE algorithm developers. Immediately after stating this, however, we referred to several reports in this special edition that document contributions to the overarching end-to-end simulation, but we did not state that the current manuscript reports on the entire end-to-end procedure and its results. What is reported on in the current paper is a select subset of the end-to-end procedure using specialized, and expensive, 3D RT simulations of "observed" radiances that were NOT used for the official end-to-end simulations (as cited).

**Reviewer**: While what I described above is end-to-end simulations of EarthCARE processes, perhaps what the authors described in this manuscript is end-to-end simulation of EarthCARE radiative closure process. If so, the difference shown in Figures 6 and 9 are caused by not knowing scenes outside of the instrument field of view that measures radiances. Is this correct? Or do cloud and aerosol retrieval errors contribute to the differences? If the retrieval errors do not contribute to the differences, the authors need to state in the manuscript.

**Reply**: The Reviewer's first comment is exactly what our paper is about. We believe we made this clear in both the "Introduction" and the "Summary and discussion" sections; see first sentence of the latter. Differences that the Reviewer refers to are indeed caused partly by errors they refer to (see our paper on the ACM-3D processor in this special edition), but also several other errors as mentioned in the paper. The Reviewer's last point is a very good one, and they are correct in assuming that we did not include retrieval errors and uncertainties (e.g., see first paragraph of section 2.3)... and for the most part, EarthCARE will not either. These quantities are remarkably difficult to define, describe quantitatively, and account for in the closure process. One advantage to using "effective fluxes", as described here, is that specification of some key uncertainties are avoided. This was a central point of the paper and was made several times.

**Reviewer**: As stated by the authors, Figures 6 and 9 show that clear-sky scenes have a smaller difference. But looking Figure 6, I cannot clearly link the smaller difference to a larger probability of closure statistics shown in Figure 6c. The authors need to link and show that smaller differences lead to a larger probability for a sanity check.

**Reply**: This is an interesting point, which we did discuss near line 445 of the original, for it draws attention to the case in which flux differences are small, perhaps much less than 10 W m$^{-2}$, but their uncertainties are smaller still resulting in the overlap of their respective distributions of possible results being (possibly very) small! While this might "appear" to a "failure", the fact remains that they differ by less than the mission target. This is where utilization of results is left up to Users to decide if they wish to use particular observations... the closure process simply reports the flux difference and the probability of that difference being less than some critical difference (e.g., 10 W m$^{-2}$).

**Reviewer**: Does the radiation closure for the ACMB-DF product use 1D radiative transfer models too? If so and if the purpose of this manuscript is to demonstrate the process, could you describe how 1D models will be used in the process?

**Reply**: Currently, 1D RT model results are not going to be used in the closure/assessment process. The express purpose for showing 1D results in this report is because 1D RTMs are still the standard for both end-to-end simulations and in most applications that involve retrievals from satellite data. Our intention here for showing differences between 1D and 3D RT model results was to highlight benefits of moving on from 1D models to the more realistic, albeit taxing, 3D RT models.

**Reviewer 2**

**Reviewer**: Fig. 1: How did you define the horizontal coordinates of the radiance image in oblique views? I am confused by the parallax shown in Fig. 1 that looks opposite to me. If the horizontal coordinates of oblique line of view is defined at the surface level, clouds in 3DRT backward image should be shifted to the right direction compared to 1DRT backward image.

**Reply**: The Reviewer is correct... this was made confusing by our not stating explicitly the reference level. The Reviewer assumed, quite reasonably, that it was the surface. In fact, it was at ~17 km (cloudtop) altitude. Had it been the surface, the Reviewer's description would be correct, but at 17 km, all clouds are at lower altitudes, so they shift opposite to the Reviewer's description (i.e., as shown in Fig. 1). This is now noted in the figure's caption.

**Reviewer**: L211-216: Then how this issue is addressed in this study?

**Reply**: We will be co-registering 3D RT model radiances using heights provided by the BMA-FLX process, which co-registers BBR radiances; these values will also be used to compute effective

fluxes. Note that solar and thermal reference levels differ in general and can be between the surface and cloudtop. For this paper, 3D RT simulations of BBR and MSI radiances were used to compute co-registration levels. As the methodology derives directly from another paper in this special issue, which has now been submitted, we have added this citation, in the paragraph in question, in the revised version.

**Reviewer**: L317: This sentence with "true" flux and Fig. 5 with "actual flux" are confusing to me. I would like to reserve the phrase "true flux" for directly observed flux, not using the ADM. Could the authors rewrite it better? At least, these fluxes should be stated as "effective" flux.

**Reply**: The Reviewer's confusion is well founded... we had used "actual" throughout in an earlier draft, but changed to "true", save for the noted occurrence in the caption of Fig. 5. We have changed "actual" to "true". Regarding the Reviewer's comment that "...“true flux" for directly observed flux...", we note that satellites never observe hemispherically-integrated "fluxes". When fluxes are ascribed to satellite measurements, they always come from an ADM, and so, by definition, are untrue.

**Reviewer**: L387, "underestimation of high ice cloud water contents.": Can this be confirmed from a comparison between initial and retrieved values of ice cloud water content? It is worth to check vertical cross section of ice cloud content in the inputs and retrievals. Can the authors show the comparison of the vertical profiles?

**Reply**: We have done that and recognize fully that this is exactly what a scientist would like to see. We discussed at length amongst ourselves whether or not to include or omit this comparison and decided to omit it because: i) it would have opened issues that would expand the paper beyond its scope; ii) its scope was to just report on the ACMB-DF process, not to go into detailed analysis of its performance, and the performance of other processes (that could be a massive paper!); and iii) many of the algorithms involved (note that the process being reported on in this paper is right at the end of the chain) were in development and their results were not ready to be scrutinized publicly; though it was felt they were in sufficiently good condition for the task at hand.

Typographic corrections:

**Reviewer**: L190, "I": should be in lowercase "i".

**Reply**: Got it... thanks.

**Reviewer**: L243, "theta0": This did not appear previously, although it appear in the Figure 1 caption. Please define it here.

**Reply**: It is defined here.

**Reviewer**: L298, "Use of": Should be "By the use of".

**Reply**: This has been altered, though not exactly as the Reviewer suggested.

**Reviewer**: L413, "Is" should be "is".

**Reply**: Got it… thanks.